# Acceptability of improved cook stoves-a scoping review of the literature

**Bipin Adhikari**[1,2]*, **Sophie Suh Young Kang**[1], **Aaryan Dahal**[1], **Salum Mshamu**[2,3], **Jacqueline Deen**[4], **Christopher Pell**[5,6,7], **Lorenz von Seidlein**[1,2], **Jakob Knudsen**[8], **Thomas Chevalier Bøjstrup**[8]

1 Mahidol Oxford Tropical Medicine Research Unit, Faculty of Tropical Medicine, Mahidol University, Bangkok, Thailand, 2 Centre for Tropical Medicine and Global Health, Nuffield Department of Medicine, University of Oxford, Oxford, United Kingdom, 3 CSK Research Solutions, Mtwara, Tanzania, 4 University of the Philippines Manila, Manila, Philippines, 5 Amsterdam University Medical Centres, Department of Global Health, University of Amsterdam, Amsterdam, the Netherlands, 6 Amsterdam Institute for Global Health and Development, Amsterdam, The Netherlands, 7 Amsterdam Public Health Research Institute, Amsterdam, the Netherlands, 8 Royal Danish Academy – Architecture, Design, Conservation, Copenhagen, Denmark

* Bipin@tropmedres.ac

**Data Availability Statement:** All data relevant to the article are within the manuscript and supplementary files.

## Abstract

Improved cooking stoves (ICS) are intended to reduce indoor air pollution and the inefficient use of fuel yet there is often reticence to shift permanently to ICS. Drawing on a scoping review, this article aims to provide a comprehensive overview of factors affecting the acceptability of ICS. A scoping review was carried out using a systematic search strategy of literature. All articles identified in three major databases that included Pubmed/Medline, Scopus and Web of Science underwent screening followed by content analysis to generate major and minor themes using a structured social level analysis. The analysis identified factors at micro, meso, and macro-social levels that potentially contribute to an adoption of an improved cooking stove (ICS). The findings from the review were discussed and refined among a group of experts identified based on their prior academic or commercial contributions related to ICS. Adoption of ICS was dependent on functional outputs (e.g. cleanliness, and cooking efficiency) while meeting local social and cultural demands (e.g. cooking large meals, traditional meals, and taste). Health and cost benefits played an important role in the adoption and sustained use of ICS. The adoption of ICS was enabled by use among neighbors and other community members. Sustained use of ICS depended on fuel supply, fuel security and policies promoting its use. Policies offering subsidies in support of supply-chain garnered institutional trust among community members and resulted in the sustained use of ICS. In addition to design attributes of ICS that could meet both scientific and social demands, policies supporting promotion of clean energy, subsidies and supplies can substantially enhance the adoption of ICS.

## Introduction

### Household air pollution and its contribution to diseases

About a third of the global population (2.3 billion) uses open fires, kerosene, biomass (wood, animal dung, and crop waste) or coal for cooking. These fuels can generate harmful household

**Funding:** This study was funded by Hanako Foundation, Singapore awarded to LvS. The funders had no role in study design, data collection and analysis, decision to publish, or preparation of the manuscript. Funding for publication was supported by Mahidol-Oxford Tropical Medicine Research Unit (MORU). The MORU is funded by the Wellcome Trust (220211/Z/20/Z). For the purpose of Open Access, the author has applied a CC BY public copyright license to any Author-Accepted Manuscript version arising from this submission. The funder had no role in the writing or preparation of the manuscript.

**Competing interests:** The authors have declared that no competing interests exist.

air pollution (HAP) [1]. HAP was responsible for an estimated 3.2 million deaths in 2020 related to the development of ischemic heart disease (32%), stroke (23%), lower respiratory tract infections (21%), chronic obstructive pulmonary diseases (19%) and lung cancer (6%). The cumulative number of deaths caused by household and ambient air pollution is estimated 6.7 million deaths per year. Children and women are the most vulnerable to HAP [2].

## Interventions addressing indoor air pollution

Multiple strategies and methods have been adopted to reduce particulate matter and other pollutants. Besides outdoor air pollution, HAP is a major contributor to morbidities and mortalities in low and middle income countries [3]. Multi-pronged interventions, such as improved ventilation, separate cooking spaces, and environmentally friendly fuels seek to minimise HAP [4]. One such intervention is improving cook-stoves.

## What is improved cooking stove?

Improved cook-stoves (ICS) bear a wide-spectrum of advances including their sources of fuel. Broadly, stoves can be categorized into 1. three-stone stoves that uses biomass fuels (for e.g. charcoal, crop residue, animal dung and wood); 2. biomass stoves produced from local low-cost materials (e.g. earthen material, and cement); 3. improved biomass stoves using newer technologies for cleanliness, better combustion and safety; 4. advanced biomass stoves that use a fan to force emission back into flame for better combustion (e.g. forced air stoves, and gasifier stoves); and 5. clean fuel stoves using clean and efficient fuels such as LPG, biogas, cooking gas, electricity and solar energy [5–8]. Fuel sources can be categorized into biomass (firewood, crop residue, animal dung, charcoal and various types of pellets), transitional fuel (kerosene) and more modern, cleaner fuels (electricity, solar energy and liquefied petroleum gas) [9]. In this article, we refer to ICS as any improvement to an existing stove, for instance that could be an improvement over a three stone stove, or modified charcoal stoves. ICS is an important indicator of global development as it is interlinked with clean energy use. Access to modern energy source for clean cooking is one of the key drivers for achieving the 2030 Agenda for sustainable development goal 7 that aims for universal access to affordable, reliable, and modern energy services and upgrading of energy technology in developing countries [10].

## What are the current literature findings around ICS acceptability?

Despite substantial progress in ICS development, the adoption of ICS remains marginal, particularly in low- and middle- income countries (LMICs). Two main concepts explain the adoption of energy options namely 'energy ladder' and 'energy stacking' [11–14]. Energy ladder explains the use of energy types (traditional to transitional to modern) proportionate to the income status of the household [15]. This concept suggests that an increase in household income will lead to a family's shift to energy sources such as liquid petroleum gas and electricity that are cleaner and more convenient to use. Energy stacking on the other hand suggests that independent of affordability, households continue to use multiple fuel options as long as multiple fuel sources generate maximum utility [11, 14, 16]. Fuel stacking is used in both rich and poorer households and depends on factors such as cash availability and cultural preference. Although these two concepts provide a broad, overarching explanation regarding adoption of ICS, a broad spectrum of factors affects the adoption of ICS on a global level. Yet reviews of the adoption of ICS, particularly referring to the development over the last decades, are scarce. The main objective of this study was to explore the factors affecting the global acceptability of ICS.

## Materials and methods

### Overview

This study utilized a narrative synthesis of literature, using a scoping review framework outlined by Arksey and O'Malley [17] and follows a PRISMA guideline for a scoping review (S1 File). Our review framework incorporates the following steps as a part of the process in knowledge translation in scoping review methodology.

### Identifying the key research questions through an iterative review/discussion

Our initial research question emerged from a randomised controlled trial of novel-design housing (Star Homes) in rural Mtwara, Tanzania. Improved cook stoves (with ventilation) were one component within the novel house design [18]. During the evaluation of Star Homes' acceptability and feasibility, the use of improved coking stoves was found to be inconsistent and heterogeneous. Some households used the new stoves all the time, others never, and a third group made use of the new stoves sometimes but fell back to using the three stone option at other times, i.e. stacking. Although social science studies have explored the use, acceptability and feasibility of novel stoves, the question of what makes a cooking stove preferable remains unanswered. This question is particularly relevant when cost is not the main driver. We based the research question on iterative discussions among the multi-disciplinary team that consisted of social scientists, architects, epidemiologists, entomologists, and field researchers [19, 20]. The overarching research question emerged as 'What factors affect the acceptability of improved cooking stoves in sub-Saharan Africa and around the globe?'

### Identifying the initial potential studies based on the discussion

Several review articles were identified that explored the use of ICS [21–29]. This work allowed us to explore niche areas specifically the adoption of ICS within the past literature including the updates that could respond to our research question.

### Searching literature in major medical databases

Three major medical databases (PubMed/Medline, Scopus, and Web of Science) were used to search all published literature until April 2024 using a set of terms in combination with Boolean operators. We used the term 'acceptability' to refer to a spectrum of agreement to use the ICS without restricting its initial adoption (S2 File).

### Collating of data, synthesizing, and reporting of the findings

Data from all three databases (n = 3405) were collated in Endnote version 21. Duplicates were removed (n = 725) followed by initial screening of articles by title and abstract (n = 2240) PRISMA flowchart (Fig 1) [30]. A total of 109 articles were retained for full text content analysis after which 20 articles were removed based on the scope or details of the content and relevance to our research question. Articles were excluded if 1. they were in languages other than English; 2. briefly mentioned ICS without adequate content on acceptability; 3. reported as comparative accounts of clean energy sources without adequate details on improved cooking stoves and 4. improved cooking stoves were assessed for their functionalities (e.g. cooking efficiency, and emission) without details on users' preference. All articles, regardless of their types with relevant and adequate content addressing our research questions were included in a full text analysis.

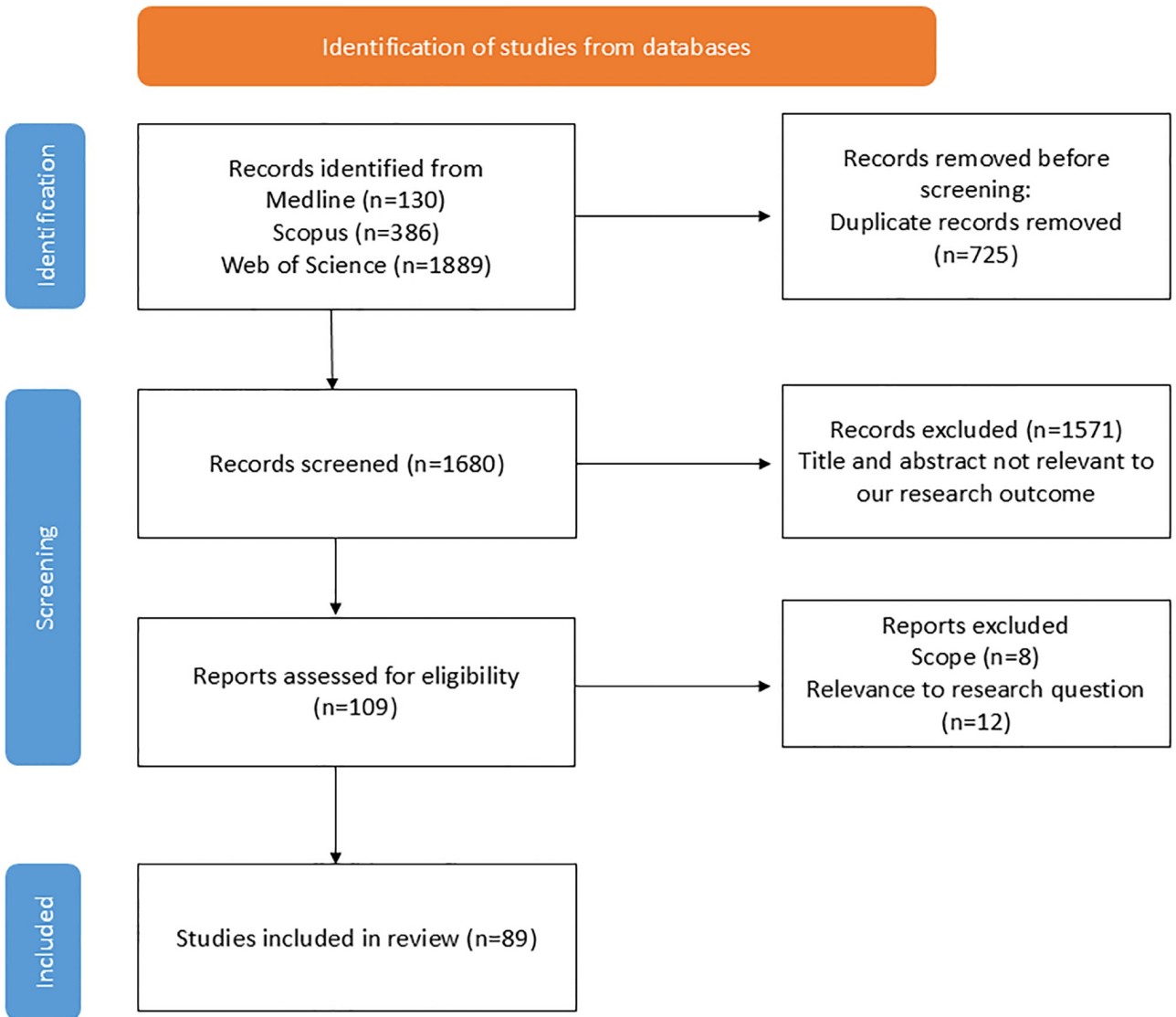

**Fig 1. Identification of structural factors contributing to the acceptability and adoption of improved cooking stove.**

All relevant data from the full text search were extracted into Microsoft Excel sheet that had thematic headings responsive to our research question. Once the data were populated into the Excel sheet, line by line reading and coding was conducted at NVivo version 14.2, a qualitative data organization software [31]. Both deductive and inductive coding was applied to create a final codebook. Based on the final layout of codes and their significance and relevance to our research question, major and minor themes were synthesized and were layered into macro, meso, and micro-social analysis frame. Using these layered approaches in social science has been deemed to offer multitude of perspectives (structural contributions) in a continuum affecting the research question [32]. The final findings were discussed among the core group of authors.

## Discussion among experts and utilizing their feedback

The thematic layout of the findings was prepared in Microsoft Power point and were discussed among the experts identified from the review. A total of three experts in the field were

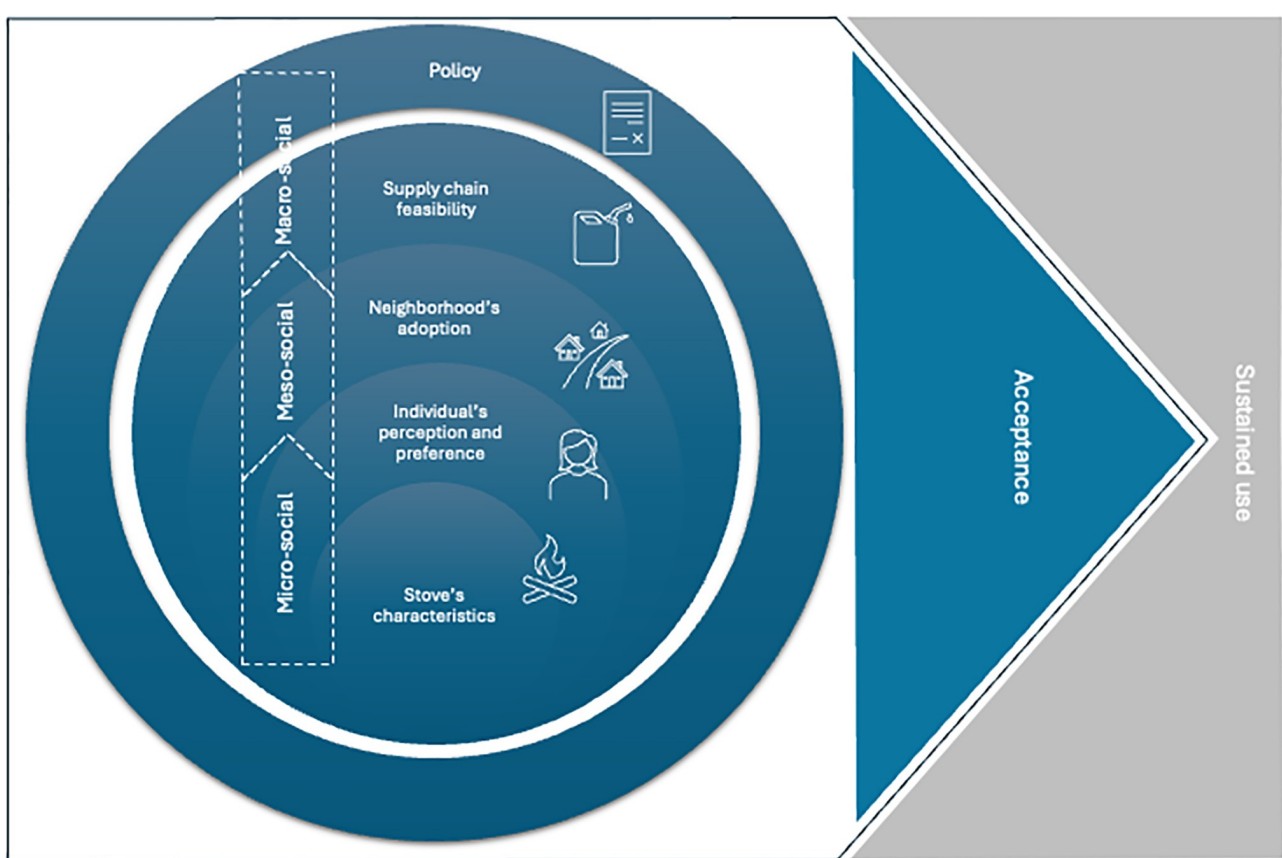

**Fig 2. Identification of structural factors contributing to the acceptability and adoption of improved cooking stove.**

approached via email and appointments were established to discuss the findings. Feedback and suggestions led to the inclusion of additional, highly relevant literature that was incorporated into the database. We identified five prominent interrelated themes that broadly ranged from micro-social to macro-social level and included: 1. Stove's characteristics; 2. Individual's perception and preference; 3. Neighborhood's adoption; 4. Supply chain feasibility and 5. Policy (Fig 2).

## Results

### Characteristics of the studies

A total of 89 studies were included in this review that consisted of nine reviews [21–29] including one systematic review in 2014 [22] and two other reviews collected literature as far back from 1980–2012 [23] and 1980–2011 [24] (S3 File). Almost all studies were conducted in LMICs, mostly in Asia, Africa and Latin America. A total of 37 studies were conducted in Africa and that included Ghana, Ethiopia, Nigeria, Tanzania, Uganda, Rwanda, Kenya, Mozambique, Malawi, South Africa, Cameroon, and Zambia [9, 12, 25, 33–66]. A total of 34 studies were conducted in Asia that included India, Pakistan, Nepal, Indonesia, Bangladesh, Vietnam, Timor-Leste, China, Cambodia, and a mixed study between Asia, Africa and America [5, 28, 67–97]. Among the 34 studies conducted in Asia, almost half were from India (n = 16). A total of 12 studies were conducted in Central/Latin America that included Peru,

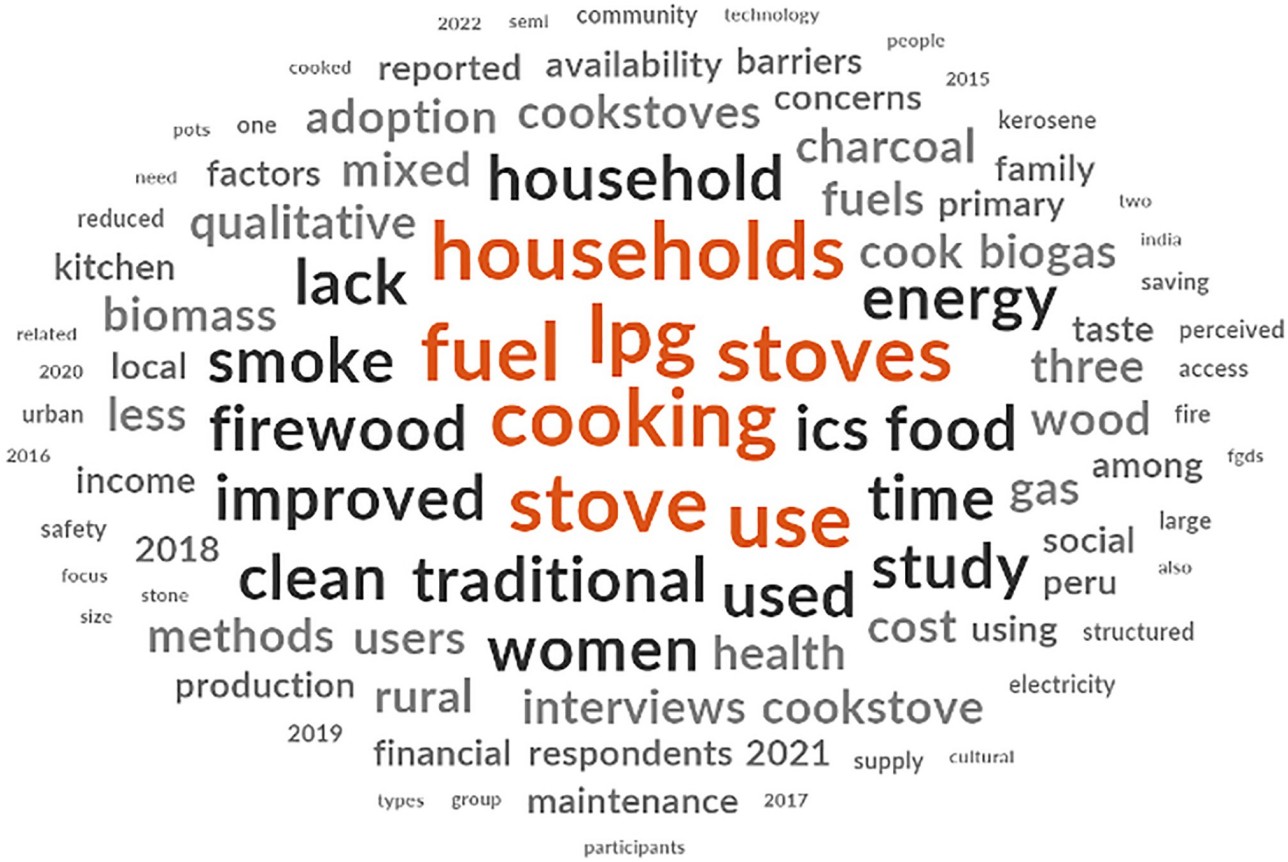

**Fig 3. Word cloud generated from NVivo to demonstrate the prominence of terms within the data table.**

Mexico, Colombia, Guatemala, Ecuador, and few were multi-country research [90, 98–108]. Most studies 81%; 72/89 included qualitative components, such as semi-structured interviews, and observations. Most sources reported studies that specifically explored the acceptability of the improved cooking stoves, others included acceptability as a component of the implementation of the improved cooking stoves.

## Stoves

**Fuel.** Studies assessed a wide spectrum of ICS and often made comparisons to traditional stoves (Fig 3). The most often mentioned ICS, LPG stoves were compared to variants of traditional stoves (three stone stoves, mud stoves, mud stoves with chimneys, kerosene stoves, clay stoves, advanced biomass stoves) [23, 24, 26, 33, 35, 36, 40, 48, 53, 55–57, 59, 62, 65–68, 70–72, 75, 79, 80, 82, 84, 87–89, 91, 94, 98, 102, 103, 105–109]. In addition, some studies also compared electric and solar powered stoves [23, 26–28, 45, 49, 52, 58, 59, 75, 81, 84]. Although the majority of studies made comparisons between ICS and traditional stoves, a large proportion of households were found to use multiple types of cooking stoves interchangeably (stacking).

While studies conducted in rural households found that most used firewood as the main fuel for cooking [12, 25, 38, 42, 43, 47, 49, 50, 52–54, 57, 67, 69, 71, 74–76, 78, 79, 82, 84, 88, 92, 95, 99–101, 103, 106, 109], semi-urban and urban households had multiple sources of cooking stoves with a predilection to use LPGs or cleaner fuels [5, 9, 25, 26, 34, 36, 40, 51, 53, 55, 56, 59,

62, 63, 68, 70, 82, 87, 94, 98, 102, 103]. In Ghana, LPG use was found to decrease gradually from 40% in the first week to less than 5% after 9 months, and over 80% of respondents continued to use their traditional three stone stoves for cooking their main meals [33].

The urban/rural divide for cooking fuel did not apply invariably. For example in some rural areas of Peru, the predominant fuel for cooking was LPG (66%) followed by wood (29%) and the remaining 5% were coal, natural gas and electricity [105]. By contrast, some urban areas in Rwanda used biomass as a predominant (80.3%) source of fuel [66]. A comparative three country-study found that in rural households, women were more likely to burn locally available biomass, such as cow, sheep, and alpaca dung and firewood in Peru; firewood, and crop residue in Kenya; and firewood, dung, crop residue and coal in Nepal [90]. Studies from India [69, 71, 73, 79, 80, 82, 109], Pakistan [78, 97], Bangladesh [72, 88, 89], and Nepal [91, 92], reported predominant use of firewood or biomass fuels, although the urban-rural divide in terms of fuel choice was increasing over the years, including the use of multiple fuel types [67, 86, 87, 95]. The stacking of fuel types including transitioning to cleaner fuel was also reported from China [75], Indonesia [5, 94] and Cambodia [5]. There was a general trend to adopt cleaner fuels globally. For instance, in Guatemala, 50.1% of urban households and 93.4% of rural households use wood for cooking, but households were gradually adopting LPG as a main source of fuel [107].

**Number of burner and size.**  A single burner LPG was generally considered inadequate to prepare meals for multiple family members [33]. A mismatch between large pots and small burners resulted in poor cooking efficiency. Particularly, when large amounts of food required prolonged and frequent stirring, an LPG stove was considered to have major constraints and thus often resorted to biomass fuels [29, 35, 39, 102]. ICS meeting demands for cooking large and traditional dishes were more acceptable [24, 65, 90]. Some studies recommended community involvement in designing community-tailored cook stoves [24]. Some LPGs with multiple burners seemingly met the demands for large number of dishes including the traditional food [63]. Cooking of large traditional food items, such as Injera in Ethiopia, Ugali or Makande in east Africa was challenging on ICS (e.g. LPGs or electric stoves), especially when preparing food for large groups during festivals [29, 39, 42, 49, 50, 73, 83, 89, 92, 93, 98, 101, 106]. In rural households, traditional stoves (e.g. with wide potholders) were better suited to cook animal feeds further contributing to the preference for traditional stoves [91]. Traditional stoves (e.g. three stone stoves and biomass fuels) were also used to warm the house during winter months, heating water for bathing, and drinking [46, 62, 92]; including to deter insects, and pests by the smoke produced [77]. Using a traditional stove was also considered by some to be an outward sign of respecting traditions and culture [109]. Durability of the stove, stability of the potholders within the stove, and portability were other factors for the adoption of novel stoves [50, 58, 64, 69, 86].

A small stove entrance for wood or other biomass was found to make the ICS unpopular among potential users in Cambodia [5]. Portability of the stove including its stability, fuelwood inlet size, stove holding attribute, size meeting demands for large family sizes also affected the preference and adoption of the ICS [38].

Specific types of ICS where top loading fuelwood were unpopular because they required frequent handling of the pots in exchange of fuel insertion and required frequent repairs [69]. Solar cookers were perceived to cook only limited amount of food items and operate under the solar light only and fail on cloudy days [52].

**Kitchen architecture.**  The available space made a major impact on type of stoves used [38, 55, 68, 97]. In urban houses without outdoor kitchens for burning firewood, an ICS was deemed essential. Even in rural households, cooking in a separate kitchen and transporting food to the main house were considered difficult, especially during the rainy season or when

members felt exhausted. This also motivated them to use ICS within the main house [55]. Having a space for kitchen inside the main house was likely to increase the adoption of ICS and cleaner fuels [97]. The use of ICS could also be affected by the indoor characteristics of the kitchen such as those with indoor separate kitchen, indoor with attached kitchen, indoor with no separate kitchen area. Traditional stoves have an additional space requirement and require the engagement of several family members, especially in rural households. The prolonged cooking time required for traditional stoves offered women and family members socialize around the cooking area, allocating a time and space for engagement not so different from the modern 'kitchen island' increasingly essential for suburban homes in high income countries (HICs) [41, 43, 72, 77, 99]. Lack of a clean energy supply chain was found to make a major impact on adoption of ICS [75]. Some studies interpreted the use of LPG as a sign of women empowerment as it meant reducing the time required for the maintenance of traditional stoves [57].

### Perception and preference of stove users

**Socio-demographics of end users.** Households headed by women (AOR 1.96; 95% CI 1.24–3.10), private house ownership (AOR 4.58; 95% CI 3.89–6.19), separate cooking location (AOR 1.84; 95% CI 1.49–2.78), and fuel purchasing capacity (AOR 2.13; 95% CI 1.64–2.76) predicted the adoption of ICS [12, 34, 41]. A lower educational level of the household head (AOR 0.31; 95% CI 0.23–0.42) predicted a lower chance of adopting an ICS [5, 25, 34, 41, 56, 68, 84, 97]. The role of decision makers in the adoption of the ICS was an important factor. For instance, women in the kitchen had to seek a decision from a male household head, a mother-in-law, or village authorities. Such decision-making dynamics affected the adoption of ICS [67]. Adopting cleaner fuel translated into belonging to a higher social class in India and thus motivated household members to adopt its use [80, 91, 92, 107]. Using firewood also implied the responsibility trickled down to other family members including children, who would have to collect firewood instead of studying [36].

**Time and cooking efficiency.** Major reasons for adopting ICS was the reduced time required and convenience of cooking particularly with LPG stoves [22, 26, 29, 33, 35, 36, 39, 42, 50, 53, 58, 60, 62, 64, 65, 69, 71, 79–81, 86, 89, 91, 92, 95, 101, 102, 104–106, 108]. Households noted a reduction in cooking time by approximately 40 minutes when LPG stoves were introduced [86]. The ICS saved resources, such as time needed to gather firewood, allowed multi-task when cooking, reduced production of smoke, increased harmony between the household members by reducing the risks of burns and other fire hazards [12, 35, 65]. It is easily overlooked that the fire and fuel of traditional stoves need constant attention, a factor that is frequently overlooked in assessments of cooking practice [36]. The durability of the stove, reduction in cooking time, decreased consumption of fuelwood, and reduced smoke production were key features that supported the preference for and adoption of ICS [38, 58, 64, 71, 74, 86]. The speed of cooking was appreciated in India, as it allowed more time for other activities, including income generation [70].

**Cost of fuel.** The main barrier to the use of the LPG stove was the availability and cost of LPG refills. For instance, the cost was found to restrain its widespread use, reserving its use for cooking food during busy hours, preparing tea, boiling water or milk for immediate use [82]. This was a major constraint in rural households who lacked adequate cash flow [23, 25, 29, 33, 44, 45, 53, 57, 105]. Community-based subsidies such as credit, and installment plans were used to promote the adoption of ICS [24]. Living in rural areas meant having poor access to the newer technology and hence favored traditional stoves [25, 73, 106]. Perceived cheap prices of cookstove (AOR 2.48; 95% CI 1.91–3.21), perceived fuel-saving benefits (AOR 1.63; 95% CI

1.18–2.24), and longer durability of cookstove technology (AOR 1.71; 95% CI 1.30–2.26) played a significant role in the adoption of ICS within the investigated households [34]. Financial constraints were the main barrier in many households [12, 25, 46, 62, 63, 67, 83, 84, 86, 87, 92]. The low or no cost of firewood was one of the barriers to adopting cleaner fuels [67, 71, 80, 109]. Saving money and fuel, as well as increasing household incomes increased the chances of switching to ICS [5, 23, 39, 47, 48]. On average, households reported a 40% reduction in wood fuel consumption after switching to clean energy such as LPG, nonetheless, the varying cost per country, access and subsidy-policy can affect it [83]. The reduced fuel consumption associated with ICS and increased awareness of ICS facilitated the adoption [37, 95].

**Health benefits.**   Smoke from biomass including the health impacts, such as cough, itchy eyes, fire related injuries and discomfort were some of the reasons why users adopted LPG stoves [9, 22, 29, 33, 49, 50, 57, 63, 66, 67, 78, 80, 90–92, 95, 98, 99, 101, 104–106]. A review reported that around 78% of the households reported a subjective decrease in eye issues such as irritation after the switch to ICS [83]. A study in India further identified the cost of illness from smoke emissions as a major incentive to adopt ICS [69]. Perceived health benefits of cook stoves (AOR 1.76; 95% CI 1.15–2.70) influenced adopting ICS [34, 37]. In contrast, lack of awareness on what household air pollution is and how it affected health were found to be a major barrier to adopting cleaner fuels [37, 41, 66, 67]. The low back pain when collecting firewood for traditional stove triggered some respondents to switch to ICS [101]. In contrast, the perception that smoke can cure food [smoke was perceived to maintain hygiene], deter insects and other pests tended to support the continued use of traditional stoves [77].

**Clean cooking.**   Cleanliness afforded by LPG stove was a prominent reason why respondents switched from traditional biomass fuels [29, 33, 36, 40, 44, 55, 61, 63, 64, 89, 100, 104, 106, 108]. One specific example was that LPG stoves did not stain pots and pans unlike the soot from traditional stoves [59, 99]. In addition, users of traditional stoves often had to remove the char from pots and pans before cooking. The walls and ceilings of rooms used for traditional cooking with open fire turned black, the air was smoky and increased the indoor temperature [9, 36, 61, 67, 79, 99]. Clean fuel combined with convenient use, easy to light, attractive design, and multiple options (e.g. options for charging) motivated users to adopt the new ICS [5, 47]. However, some respondents preferred outdoor cooking, and others avoided ICS because of lack of familiarity [60, 69]. There was a limited awareness on solar cookers but the perception that such cookers were unaffordable, difficult to store, and require constant sun shine will make their introduction cumbersome [52].

**Taste of food.**   Those who preferred food cooked on traditional stoves reported that the taste of food prepared on firewood stoves tasted better than food cooked on ICS [12, 22, 33, 69, 72, 80, 86, 92, 98, 100, 102]. The food cooked on LPG stoves was perceived to be 'smelly' due to the exposure of the gas, and was perceived to be 'unnatural' [26, 88, 96, 102, 103], or even impure and contaminated [96]. For instance in India *Chapatis* was perceived to be tastier when cooked in firewood [73]. Several studies reported that end users strongly perceived the food cooked in firewood as smoky and did not taste as good as food prepared on an LPG stove [35, 36, 68]. Overall, there was no consensus whether ICS prepared food was more or less tasty than food prepared on a traditional stove [23].

## Conformity

Influence from neighborhoods, family members, and friends had a major impact on the uptake of ICS [29, 40, 50, 61, 108, 109]. In an interventional study to explore the adoption of LPG, rumors and misconceptions negatively affected the adoption [35]. Optimistic previous social interaction (AOR 1.81; 95% CI 1.46–2.26), traditionally suitable (AOR 1.58; 95% CI 1.28–

1.95), and live demonstrations (AOR 2.47; 95% CI 1.98–3.07) all had a positive influence on adoption of the ICS [34]. More than income, conformity with social norms was found to influence the use of the type of cooking stoves. Some of these social behaviors (energy selection) were to a large degree dependent on culture, and social trust [68]. The opinions of family members, neighbors, and friends including public communication increased the adoption of the ICS [68, 69]. Individualized one to one training increased ICS adoption compared to group-based training [51]. Connection to a network of ICS users promoted the early and sustained adoption of ICS [51].

**Information and role of the media.** News and media messages were found to have a major impact on the adoption and sustained use of fuel for cooking [93]. For example, sensationalized reports of LPG explosion, safety concerns and associated risks were likely to trigger undue fear leading to hesitation in using LPG stoves [29, 59, 68, 79, 88, 102, 106, 108]. Similarly, not receiving adequate information about LPG could also prevent users from understanding the benefits of LPG [24, 40, 79, 88, 93, 95]. Constructive and positive information dissemination about the cleaner energy and ICS, including training were likely to promote the acceptability [24, 26, 66]. Information on the impact of clean energy use, environmental impacts (deforestation), were likely to promote the use of clean energy and ICS [26, 28, 66, 96].

## Supply-chain feasibility

**Supplies.** The limited availability of LPG required travel to refill the cylinders, the direct and indirect costs associated with travel, and safety concerns were likely to discourage the adoption of ICS [22–26, 40, 42, 45, 47, 48, 50, 53, 56, 63, 71, 87, 94, 97, 98, 101, 103, 106]. Easy availability of cookstove (AOR 1.81; 95% CI 1.5–12.17) made adoption more likely [34, 39]. In India, The Pradhan Mantri Ujjwala Yojana (PMUY) distributed over 80 million new LPG stoves across the country between 2016 and 2019 and well over 100 million new LPG stoves in 2023 [110]. Under this continuing national subsidy-scheme, consumers pay for the stove and the cylinder deposit and reduced the upfront cost. This has increased the access of LPG to over 95% of households [70]. In Nepal, availability was found to be compromised by limited energy infrastructure, including poor and intermittent access to LPG refills, and affected the adoption and use [76]. A study from Uganda highlighted the feasibility and benefit of building the ICS locally through the provision of training to community members, and use of locally available resources which could resolve the supply and dependence issues [43]. Lack of coordination, poor regulation, and insufficient market development were found to affect the supply of ICS negatively [34].

Regulation of fuel supply (market lock-in) in Indonesia, had an effect on the purchase and adoption of fuel for cooking [68]. Complex refilling processes, running out of gas, and delivery difficulties affected the preference and adoption of ICS [70]. Long distance travel requirements to a market where an ICS was available was a barrier to adopt the stove [37, 40, 98]. The shortages of LPG cylinders, incomplete filling of the cylinder, frequent price increases and cylinder leakages were listed as major disincentives against the adoption of LPG [40]. Adequate access to forest resources nearby and alternative fuel sources (e.g. dung,) also prevented adopting ICS [12, 37, 41, 43, 72, 85, 92].

**Stacking.** Some studies demonstrated that the users wanted to stack alternative fuels as a back-up [33, 62, 74, 92, 94]. But stacking was also a barrier to the long-term adoption of ICS [23]. The reviewed studies indicated that stoves were stacked for four main reasons: 1) time saving from parallel cooking on multiple stoves, 2) inability of the primary cookstove to cook all dishes, 3) housing arrangements that preclude use of certain fuel types and lastly 4) availability and cost [55, 82]. Lack of fuel, or the expectation of fuel shortages motivated stacking in

anticipation [5, 9, 71]. Livestock and land holding also motivated users to stack fuel sources as alternate energy sources for cooking [38]. Bigger family size demanding large quantities of food meant that respondents stacked multiple stoves for security [69].

### Policy

Free LPG packages and subsidy transfer attracted people to use it in preference to kerosene as their main cooking fuel. Government backed subsidies, the community's trust in fuel policy, and institutional reputation affected fuel choices [24, 28, 59, 68, 69, 91]. Government's initiatives on cleaner energy, including the dissemination of information related to the benefits of clean energy was demonstrated to make a major impact on adoption of ICS in rural China [75]. India's PMUY offers another example on how a government subsidy scheme enhanced the use of cleaner fuel (LPG) increasing the nationwide access [70, 110]. A lack of interaction between implementers and users, lack of financial subsidies for ICS acquisition, and the absence of policies in support of ICS market development negatively affected the adoption of ICS [104]. Government-led information, communication, and education initiatives to highlight the role of the household in indoor air pollution related ill-health was effective in promoting the use of ICS in Nepal [76].

Energy policy to support LPG use in Indonesia, which entailed free starter packages followed by subsidized supply of LPG at a lower price than kerosene, was found to promote the use of LPG. The transition to LPG was driven by Indonesia's fuel subsidy policies (Zero-kero Programme achieved a fivefold increase in domestic LPG consumption) [94]. Associated with the transitional policy, the use of firewood was halved over the same period. Inadequate institutional support and technical assistance had a negative effect on the implementation of energy infrastructure transition [46].

Policy awareness on how fuelwood contributes to deforestation and indoor air pollution played a critical role in the advocacy for cleaner energy use in cooking [77]. Joint efforts by the government and major stakeholders such as WHO, USAID, and NGOs promoting smokeless energy (stove) was deemed critical for success [77].

### Discussion

The adoption of ICS was contingent upon their ability to meet functional requirements, such as efficiency and cleanliness, while also aligning with social and cultural expectations. The health and economic benefits associated with ICS had a significant role in their uptake. The adoption of ICS within a community was further influenced by their use among neighbors and other community members. Effective supply chain management and the implementation of supportive policies were found critical to ensure the sustained utilization of ICS. Adoption was likely to be higher when policies provided subsidies including uninterrupted supply chain and was likely to promote institutional trust.

Two broad concepts were described a) the energy ladder that proposes use of energy to be dependent on economic caliber and b) stacking fuel for security purposes [11]. In practice both approaches are operating in concerted fashion ultimately promoting adoption of multiple sources of fuels (Fig 4). Adopting an ICS is a complex social process and operate at multiple levels from individual to institutional and policy within the context [111, 112]. Nonetheless, several theories aiming at individual behavioural interventions are limited by their scope as acceptability and adoption of ICS are inevitably shaped by broader context of social, and policy structures [111]. In a wider social and political context, acceptability of an improved cook stove is thus a complex social process echoing the elements of 'theory of adoption' and 'innovation diffusion process'. Both entail development and commercialization of an innovation,

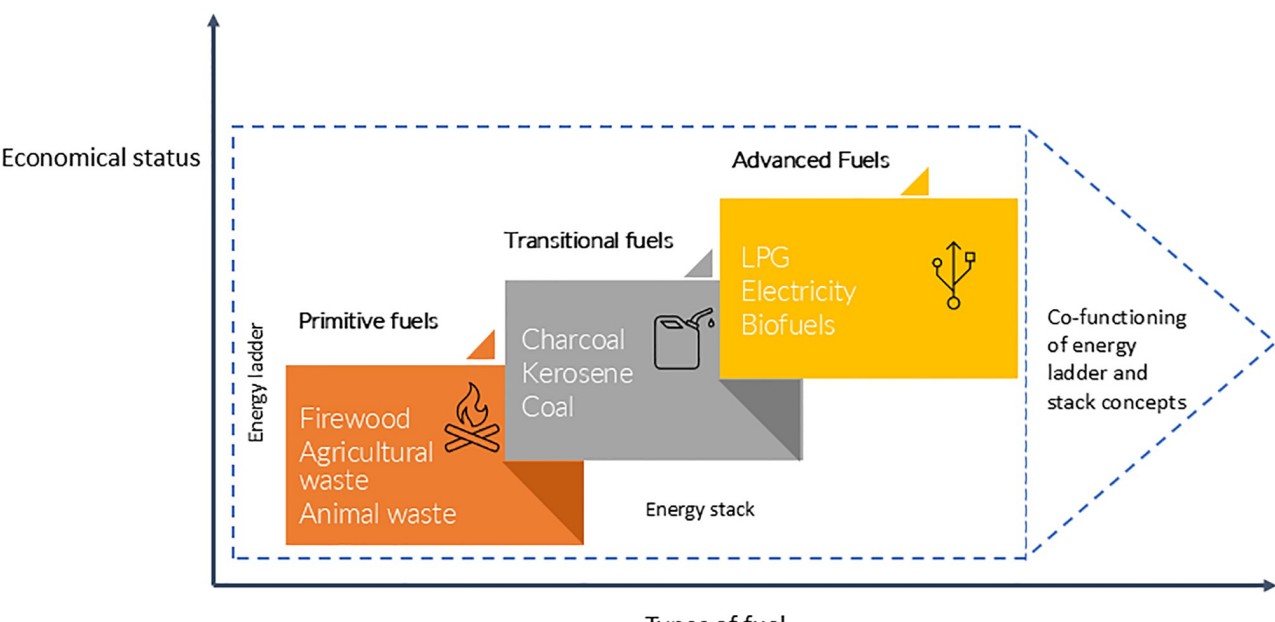

**Fig 4. Co-functioning of energy ladder and stack concepts based on the review.**

followed by the diffusion and adoption of the innovation by the users and its consequences including the critical roles of policy and social support [5, 113]. Five stages of the innovation decision process involve steps of disseminating knowledge and persuasion to decision making followed by implementation and confirmation (practiced through reflection on whether or not to sustain the use) [12, 113]. The decision-making steps when using novel equipment or technology are also echoed across the discipline. For instance, adoption of a new product in medical sciences and health care are regulated by WHO's policy on biomedical products that are assessed based on their availability, accessibility, appropriateness, and affordability [114, 115].

Cleaner fuels are increasingly used but their adoption depends on the functionality of the stove, cooking capacity, ease of use based on their designs, the required size for pots and pans, family or group size to be catered for and traditional food items that are thought to require firewood [8, 29, 35, 39, 102]. The choice and use of ICS was also dependent on the nature of kitchen that varied between urban to rural regions.

End-users were the ultimate decision makers of ICS, but their socio-demographic characteristics such as their educational level, economic caliber, and decision-making power at household level including the role of gender affected the adoption of an ICS [9, 12, 34, 41, 81]. Although women are often responsible for household tasks, including kitchen duties and cooking, their role in deciding to use ICS was limited, with the decision frequently made by the male income earner, particularly in the context of cultural patriarchy, and access to financial resources [67, 77, 83]. This suggests the need to design interventions incorporating gender and equity issues in LMICs to give women agency in choosing their preferred stove [83, 116]. An individual was likely to choose the type of stove based on the time consumed by the given stove, the quantity of food required, the specific dish (e.g. traditional dishes), social aspects (kitchen as a hub for socialization in a house) and finally the cultural demands of a typical kitchen (taste of food, and alignment with tradition) [41, 43, 72, 77, 99]. The health benefits of

ICS and cleaner energy were increasingly recognized and the transition to cleaner energy seemed to be driven largely by the health benefits (or health hazards by the firewood) [27].

Household members, neighbors, community members, and wider social circle play a subtle yet important role in adopting innovative technologies including ICS and cleaner energy. This aligns with the principles of diffusion theory on how a majority of individuals (for example, early and late majority) are influenced by the early adopters while others can end up as laggards or non-adopters [113, 117].

The diffusion theory explains how malleable an individual's decision can be, where constructive information and positive feedback promote the adoption [118]. The role of media in promoting the benefits of ICS, cleaner energy, including health benefits can be supportive for the adoption and sustained use of new stoves [119]. Negative media coverage has certainly a disruptive effect on ICS use. This is also where community and public engagement are found to be critical, as it can leverage the institutional trust and sustain the use of ICS [120–122].

Government policies regulating supplies and promotion of the energy sources can have significant impact on the uptake of ICS [123, 124]. Availability, accessibility, and affordability are fundamental contributors in adopting a new technology, and thus any impediments to these critical elements affect the adoption and sustained use of ICS [114, 115]. Policy engagement on the need of clean energy such as subsidies had a major impact on the adoption and sustained use of LPG stoves [70, 75, 94, 110]. Government's initiatives on cleaner energy, had a significant impact on adoption of ICS in rural China [75]. India's The Pradhan Mantri Ujjwala Yojana (PMUY) increased the access to LPGs to more than 100 million households [70, 110] and Indonesia's 'Zero-kero Programme' achieved a fivefold increase in LPG consumption [94]. Achieving nationwide use and sustainable adoption of LPG relies on sustained governmental initiatives to ensure accessibility, financial support, and active engagement with the population [125]. Despite government's aim to promote cleaner energy (LPG and electricity), 80% of population in rural Nepal still use solid-biomass fuels, partly because of the absence of consistent access to LPGs and subsidy scheme by the government [126, 127]. The rural versus urban disparity in use of LPG was apparent in many countries [23, 55, 57, 87, 105]. For instance, in Peru, the main barrier to adopting LPG was the availability and access to the LPG in rural villages [103]; and thus requires a focused and tailored approach in ensuring the access, and subsidies. A recent review echoes how government's policy initiatives are critical to ensure the access, affordability and adoption of LPGs [116].

As a part of the review, discussions with the experts offered us some practical insights on how the LPG stoves were utilized in high income countries. For instance, experts highlighted the supply of piped LPG to the kitchen in Europe in response to the safety issues with use of LPG cylinders in the past. Experts also referred to cleaner energy alternatives such as electric cooking as the gold standard for the future. The sustainability of LPG, a fossil fuel, was thought to require re-consideration for future. Considering these practicalities of energy supply and its association with environment will be critical for the future of ICS. Designing and modifying the kitchen's architecture for better ventilation was considered an immediate and cost-effective step to optimize the health benefits of ICS to populations in the LMICs.

## Strengths and limitations

This was a scoping review exploring the acceptability of improved cook stoves and included literature from around the globe. By virtue of the topic, and the wide scope, the heterogeneity of literature spanned across various disciplines (marketing research, implementation, health

sciences, architecture, and energy science). This review had to integrate a range of inter-disciplinary perspectives. As a scoping review, reiterative discussions were held among internal experts and external experts and their feedback were integrated into the manuscript. By its nature, the review could not quantify the adoption of ICS globally. The review is limited by the language restrictions, as we only included English language literature. The term 'ICS' used in this review specifically meant switching of traditional stove to any advanced categories, this may have generalized the information related to specific stove categories. In addition, because bulk of the literature reported the LPG stove as the ICS, our analysis on acceptability was focused on LPG stove and may have generalized the findings. Although this review explored factors affecting acceptability of ICS operating at various levels, future research could explore more on how specific types of ICS have varying impacts on adoption and sustained use over time. While initial adoption and sustained use of ICS are often not delineated in the literature, in part because these issues often co-exist as explained by the stacking of multiple energy sources, disentangling factors affecting stacking versus use of specific ICS and type of fuel could highlight the drivers of ICS and fuel types. In future, policy analysis could further highlight the impact on acceptability of ICS.

## Conclusions

This scoping review identified the factors underpinning acceptability of improved cooking stoves at multiple levels. Adoption of ICS was dependent on its functional outputs (e.g. cleanliness, and efficiency), meeting the local social and cultural standards and expectations (e.g. cooking large meals, traditional meals, and taste) and the health and cost benefits. The adoption of ICS was influenced by its use/adoption by neighbors, and wider community members. The continued use of ICS relied on the supply-chain as well as policy regulating and promoting its use. Policies offering subsidies, facilitating uninterrupted supply-chain promoted institutional trust among community members which ultimately enabled sustained use of ICS. The findings demonstrate the multitude of factors affecting the adoption of ICS and thus adapting the cooking context (e.g. kitchen design) with better ventilation may be one of the imminent solutions to impart health benefits.

## Supporting information

**S1 File. PRISMA checklist.**
(PDF)

**S2 File. Search strategy and flowchart showing article selection process.**
(DOCX)

**S3 File. Table showing final selected articles and thematic extraction of data.**
(XLSX)

## Acknowledgments

We are grateful to Nipun Shrestha for his support and contributions to design search strategies and initial data screening. We would like to express our gratitude to experts: Eva Rehfuess, a principal reviewer of this topic that has been published and referenced in our current review. We are grateful to our colleagues: Arjen Dondorp and Marja Schilstra Dondorp, and colleagues from African Clean Energy: Ruben Walker and Alice Troostwijk for their support with the review discussion and feedback.

## Author Contributions

**Conceptualization:** Bipin Adhikari, Salum Mshamu, Jacqueline Deen, Christopher Pell, Lorenz von Seidlein, Jakob Knudsen, Thomas Chevalier Bøjstrup.

**Data curation:** Bipin Adhikari, Sophie Suh Young Kang, Aaryan Dahal.

**Formal analysis:** Bipin Adhikari, Sophie Suh Young Kang, Aaryan Dahal.

**Methodology:** Bipin Adhikari, Sophie Suh Young Kang, Aaryan Dahal, Salum Mshamu, Jacqueline Deen, Christopher Pell, Lorenz von Seidlein, Jakob Knudsen, Thomas Chevalier Bøjstrup.

**Resources:** Bipin Adhikari.

**Software:** Bipin Adhikari.

**Supervision:** Bipin Adhikari, Lorenz von Seidlein, Jakob Knudsen.

**Validation:** Jakob Knudsen, Thomas Chevalier Bøjstrup.

**Visualization:** Bipin Adhikari.

**Writing – original draft:** Bipin Adhikari.

**Writing – review & editing:** Bipin Adhikari, Sophie Suh Young Kang, Aaryan Dahal, Salum Mshamu, Jacqueline Deen, Christopher Pell, Lorenz von Seidlein, Jakob Knudsen, Thomas Chevalier Bøjstrup.

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
