## [Decision Letter · Decision Letter 0]

23 Oct 2024

PGPH-D-24-01963

Acceptability of improved cook stoves--a scoping review of the literature

Dear Dr. Adhikari,

Thank you for submitting your manuscript to PLOS Global Public Health. After careful consideration, we feel that it has merit but does not fully meet PLOS Global Public Health’s publication criteria as it currently stands. Therefore, we invite you to submit a revised version of the manuscript that addresses the points raised during the review process.

We look forward to receiving your revised manuscript.

Kind regards,

Naveen Puttaswamy, Ph.D

Academic Editor

Journal Requirements:

Additional Editor Comments (if provided):

Reviewers' comments:

Reviewer's Responses to Questions

**Comments to the Author**

1. Does this manuscript meet PLOS Global Public Health’s publication criteria? Is the manuscript technically sound, and do the data support the conclusions? The manuscript must describe methodologically and ethically rigorous research with conclusions that are appropriately drawn based on the data presented.

Reviewer #1: Yes

Reviewer #2: Partly

2. Has the statistical analysis been performed appropriately and rigorously?

Reviewer #1: I don't know

Reviewer #2: N/A

3. Have the authors made all data underlying the findings in their manuscript fully available (please refer to the Data Availability Statement at the start of the manuscript PDF file)?

Reviewer #1: Yes

Reviewer #2: Yes

4. Is the manuscript presented in an intelligible fashion and written in standard English?

Reviewer #1: Yes

Reviewer #2: Yes

5. Review Comments to the Author

Reviewer #1: The title is concise and research question is clearly identified. The main elements of the topic are covered, however few topics need to be elaborated as presented in the comments. The number of studies identified and selected for the scoping review is discussed and clearly detailed in the flow chart (figure 1). In the figure 1, the reason for records excluded (n=1671) may be briefly mentioned as specified in other boxes. The authors shall identify gaps and future areas of research especially in the ICS context.

Interventions addressing indoor air pollution

1. Multi-pronged interventions, such as improved ventilation, separate cooking

spaces, and environmentally friendly fuels seek to minimise HAP [4].

Comment: Do the reference cited only presented the three types of interventions or if the authors have not discussed about the ICS?

What is improved cooking stove?

2. 2. biomass stoves produced from local low-cost materials (e.g. cement stove);

Comments: Elsewhere in the world, especially in India, stoves are also constructed with earthen material. The authors shall clarify by including this.

3. 3. improved biomass stoves using newer technologies for cleanliness and safety;

Comment: Is new technologies are for cleanliness and safety? It shall be better combustion as well. Please clarify

4. Fuel sources can be categorized into biomass (firewood, crop residue, animal dung, and charcoal), transitional fuel (kerosene) and more modern, cleaner fuels (electricity, solar energy and liquefied petroleum gas) [6].

Comment: Do the author agree that there are biomass pellet/processed type of fuel? I see the literature review have not captured this. Please clarify.

5. Comment: In the ICS section, the authors have not discussed about the types of improved cook stoves. The article DOI: 10.1007/s10393-014-0976-1 and 10.1016/j.aogh.2015.08.009 presents about testing various types of commercial stoves. Similarly, there should be other types of stoves elsewhere (eg. rocket stoves). Additional literatures shall be looked into it. Please clarify.

What are the current literature findings around ICS acceptability?

6. Two dominant theories explain the adoption of energy options namely ‘energy ladder’ and ‘energy stacking’ [8-11]

Comment: Can theories be reworded? Later in the discussion the authors have used “two broader approaches were described a) the energy ladder that proposes use of energy to be dependent on economic caliber and b) stacking fuel for security purposes [8]. Theory might mean a different aspect. Can this be reworded?

Fuel

7. Comment: In the last paragraph, some narrations on fuel use in other countries shall be included.

8. Comment: Preferences of fuel use is not discussed in this section. People generally use biomass stove for animal fodder preparation, water boiling for bathing and large meal preparation. The author shall narrate on this.

Kitchen architecture

9. Comment: There are different types of kitchen configuration. For example, outdoor cooking, indoor with separate kitchen, indoor with attached kitchen, indoor with no separate kitchen area etc. Several literatures have been published on this. The authors shall narrate on the implication of different types of kitchen on ICS use.

Cost of fuel

10. On average, using clean energy such as LPG meant households reported a 30% reduction in fuel expenses [24].

Comment: Are the authors sure that the cost of LPG would be same across different countries? This particular reference may show 30%. Due to the varying cost the reduction in fuel expenses may vary. Emission reduction percentage could be accepted as this is not going to be different irrespective of the location of combustion. Still it may vary due to the variations in the composition of LPG in different countries. Pleas clarify with some changes.

Supply-chain feasibility

11. In India, 45% of rural households were found to face difficulties in obtaining regular LPG cylinder refills [24].

Comment: In India through the Pradhan Mantri Ujjwala Yojana (PMUY) national scheme, nearly 100 million free LPG stoves and new connections are provided. Moreover, the LPG distributor network is well established for more than several decades. Hence, there should no difficulties in obtaining cylinder refills. There should be other reason such as illegal market or underground economy. Recent references will help better understand the situation. The author shall clarify this.

Policy

12. While the policies in few countries are reviewed, the Pradhan Mantri Ujjwala Yojana (PMUY) national scheme for providing free LPG stove and connections in India is not reviewed. Please clarify.

General comments:

13. This scoping review largely have focused on LPH as ICS, other types of Biomass ICS have not been reviewed. ICS includes all categories of stoves and fuel. This review looks biased toward one type of stove. The author shall clarify this.

14. The author rightly mentioned the barrier to use LPG stove was availability and the cost in the “Cost of fuel” section.

Comment: In this or in the staking section, I have not observed any literatures discussing the purpose or preferences of using LPG. Generally due to the high cost, people use LPG to boil water, milk or heating that required immediate use rather than using biomass stove. This has not been discussed in this review. Please clarify.

15. The author has not reviewed any emission standard for the ICS. The author shall narrate on this. Please clarify.

16. Reduction in pollution concentration or exposures could also be a factor to switch over to ICS. Many ICS studies are published. The authors shall include a section on this as this will be within the scope of research question. Please clarify.

Reviewer #2: Thank you for the submission. While the topic is important and timely, I have some major concerns:

1. While the manuscript mentions an overarching goal to explore the factors affecting ICS acceptability, it does not clarify how these factors will be systematically examined or which specific outcomes (e.g., health, economic, or social impacts) are being prioritized. Additionally, it does not differentiate between initial adoption and sustained use.

2. The manuscript relies heavily on qualitative themes, though quantitative evaluations exist, especially on health outcomes and cost savings. Integrating both qualitative and quantitative evidence would strengthen the findings.

3. Adoption factors are described but not sufficiently grounded in theoretical frameworks. While the “energy ladder” and “energy stacking” theories are mentioned, these could be explored in more depth. Incorporating frameworks like the Technology Acceptance Model and COM-B could enhance coherence in discussing adoption behaviors.

4. While the manuscript includes studies from Africa, Asia, and South America, it lacks regional comparisons. Comparing regional adoption rates, barriers, and enablers would improve understanding of how context shapes ICS acceptability.

5. The methodology lacks transparency on study inclusion/exclusion.

6. The policy recommendations are too broad. More context-specific recommendations with evidence on the most effective policies or subsidies would make the conclusions more actionable.

7. The manuscript does not adequately address sustainability challenges, such as scaling up LPG.

8. Gender and equity issues are not adequately addressed. How gender dynamics, income inequality, and financial access affect ICS adoption could be discussed in more depth.

9. More contextualization is needed, e.g., are the barriers to adoption different in urban vs. rural settings?

10. The manuscript could better emphasize research gaps and suggest future research directions.

6. PLOS authors have the option to publish the peer review history of their article (what does this mean?). If published, this will include your full peer review and any attached files.

**Do you want your identity to be public for this peer review?** For information about this choice, including consent withdrawal, please see our Privacy Policy.

Reviewer #1: **Yes: **Sankar Sambandam

Reviewer #2: **Yes: **Rachit Sharma

---

## [Editor Report · Decision Letter 1]

26 Nov 2024

Acceptability of improved cook stoves--a scoping review of the literature

PGPH-D-24-01963R1

Dear Dr. Adhikari, 

We are pleased to inform you that your manuscript 'Acceptability of improved cook stoves--a scoping review of the literature' has been provisionally accepted for publication in PLOS Global Public Health.

Best regards,

Naveen Puttaswamy, Ph.D

Academic Editor

Dear Dr Adhikari

Thank you for taking the time to address all comments of the reviewers. Also, thank you for your patience while awaiting the editorial decision.